# Role of Endoscopic Techniques in the Diagnosis of Complications of Allogeneic Hematopoietic Stem Cell Transplantation: A Review of the Literature

**DOI:** 10.3390/jcm13154343

**Published:** 2024-07-25

**Authors:** Ayrton Bangolo, Shraboni Dey, Vignesh Krishnan Nagesh, Kabir Gumer, Lida Avetisyan, Saima Islam, Monika Sahotra, Melissa Millett, Budoor Alqinai, Silvanna Pender, Yazmika Dunraj, Habiba Syeda, Beegum Tasneem, Mikel Duran, Nicoleta De Deugd, Prasad Thakur, Simcha Weissman, Christina Cho

**Affiliations:** 1Department of Internal Medicine, HMH Palisades Medical Center, North Bergen, NJ 07047, USA; ayrton.bangolo@hmhn.org (A.B.);; 2Division of Bone Marrow Transplant and Cellular Therapy, John Theurer Cancer Center, Hackensack, NJ 07601, USA

**Keywords:** allogeneic stem cell transplantation (Allo-SCT), graft-versus-host disease (GVHD), esophagogastroduodenoscopy (EGD), flexible sigmoidoscopy (FS), capsule endoscopy, CMV esophagitis/colitis, hepatic sinusoidal obstruction syndrome

## Abstract

Allogeneic stem cell transplantation (Allo-SCT) implies that a donor and a recipient are not genetically identical. Allo-SCT is used to cure a variety of conditions, including hematologic malignancies using the graft versus tumor effect, nonmalignant hematologic, immune deficiencies, and, more recently, genetic disorders and inborn errors of metabolism. Given the immunosuppressive and myeloablative nature of some of the conditioning chemotherapy regimens used during the Allo-SCT, patients are often at high risk of infection, including viral infections affecting the gastrointestinal tract, following the transplant. Furthermore, other complications such as hepatic sinusoidal obstruction syndrome (SOS) or graft-versus-host disease may occur post-transplant and may require endoscopy to assist in the diagnosis. This review will provide newer insights into the importance of endoscopic techniques in the diagnosis of post-Allo-SCT complications with a focus on safety and timing.

## 1. Introduction

Hematopoietic stem cell transplantation (HSCT), which was introduced in the mid-20th century, has since been used as a potential cure for nonmalignant lymphohematopoietic disorders, myeloid and lymphoid malignancies, immune deficiencies, genetic disorders, and inborn errors of metabolism [1]. HSCT can be described according to the relationship between the patient and the donor and by the anatomic source of stem cells [2]. When the donor and the recipient are not genetically identical, the HSCT is termed allogeneic [2]. 

Graft-versus-host disease (GVHD) is a potential complication of allogeneic stem cell transplant (Allo-SCT), in which immune cells transplanted from a non-identical donor (the graft) recognize the transplant recipient (the host) as foreign, thereby initiating an immune reaction that causes disease in the transplant recipient [3]. GVHD mainly affects the gastrointestinal tract (GIT), the skin, and the liver [3]. Although GVHD diagnosis can be made clinically, some cases may require the use of either combined or single use of esophagogastroduodenoscopy (EGD) and/or flexible sigmoidoscopy (FS) [4]. 

Viral infections, such as cytomegalovirus (CMV) infection, also remain a major cause of morbidity and mortality in patients having received Allo-SCT despite the use of antiviral prophylaxis [5,6]. Like GVHD, CMV infection can affect the gastrointestinal tract (GIT), causing esophagitis and/or colitis. Since symptoms of CMV esophagitis/colitis cannot be told apart from GVHD, an endoscopy with tissue biopsy is often warranted to help guide management [7]. Hepatic sinusoidal obstruction syndrome (SOS), which is characterized by injury to sinusoidal endothelial cells, is a systemic endothelial disease that typically presents within the days or weeks after Allo-SCT [8,9]. The rupture of esophageal varices in post-transplant patients with hepatic SOS has been reported in the literature [10]. Likewise, upper GI bleeding has been reported as a result of vascular ectasia in post-transplant patients [11]. 

This review will provide a concise and up to date overview of the impact of different endoscopic techniques in the diagnosis and management of complications of Allo-SCT. Furthermore, we will focus on the timing of the endoscopy and the safety of such techniques in post-transplant patients.

## 2. Complications of Allo-SCT and the Use of Endoscopic Techniques

### 2.1. GVHD

#### 2.1.1. Acute GVHD

Acute graft-versus-host disease (aGVHD) is a multisystem disorder commonly encountered in allogeneic transplanted patients caused by the donor’s immune cells reacting against the host that occurs within 100 days following transplant [12]. aGVHD presents with a classic maculopapular rash, abdominal cramps, persistent nausea and/or emesis, green, mucoid, or watery diarrhea, and a rising serum bilirubin concentration [12]. Clinically significant acute GVHD occurs in patients who receive an allogeneic hematopoietic cell transplant (HCT) despite intensive prophylaxis with immunosuppressive agents. Up to 50 percent of patients with a genotypically human leukocyte antigen (HLA)-identical sibling have been reported to develop aGVHD; matched unrelated donors and haploidentical-related donors are also risk factors for aGVHD [13]. The severity or incidence of aGVHD is not affected by whether the graft originated from the peripheral blood or the bone marrow. Increasing age, concomitant CMV of the host of the donor, or Epstein–Barr virus (EBV) seropositivity of the donor can play a role in the incidence of aGVHD [14,15,16]. 

The diagnosis of aGVHD is often clinical and can be made in patients with classic symptoms (rash, diarrhea, etc.) starting within 100 days of Allo-SCT. However, the classic symptoms of aGVHD can overlap with many isolated conditions, i.e., diarrhea can be caused by infectious colitis, a skin rash can be caused by medication use [17]. A recent study by Terdiman et al. [18] demonstrated the importance of endoscopic small-bowel biopsy in patients with upper GI symptoms following Allo-SCT. The study by Cloutrier et al. [19] concluded that upper and lower endoscopy had a similar diagnostic yield in patients with known or suspected GVHD involving the gut. However, they preferred the use of EGD over lower endoscopy given its ease and safety profile [19]. A study by the University of Stanford demonstrated that sigmoidoscopy alone had an equivalent diagnostic yield for aGVHD compared with the addition of EGD or performance of full colonoscopy [20]. Thompson et al. [21] demonstrated that colonoscopy and ileoscopy or flexible sigmoidoscopy plus upper endoscopy had the highest diagnostic yields for aGVHD. A study carried out at MD Anderson concluded that biopsy of the rectosigmoid colon is the single best test for diagnosing GI GVHD [22]. 

A more recent single-institutional study suggested that combined EGD and FS with biopsy of the stomach and rectosigmoid colon results in the greatest diagnostic yield for most patients with aGVHD, independent of symptoms [4]. They confirmed that biopsy of normal-appearing mucosa is very important, as endoscopic evidence of aGVHD usually appears in severe aGVHD. Furthermore, the study found no significant difference in adverse events between thrombocytopenic and neutropenic patients, confirming the safety of endoscopy in this patient population [4]. 

Based on this literature review, we propose using FS in patients with suspected aGVHD with diarrhea as the predominant symptom. EGD can be considered if the patient’s presentation predominantly involves the upper GIT, i.e., severe nausea with multiple episodes of vomiting. FS/EGD should be performed as soon as the diagnosis of aGVHD is suspected, and thrombocytopenia and/or neutropenia should not be contraindications. Even in the presence of normal mucosa during FS/EGD, a sample should be taken for biopsy as evidence of aGVHD can be seen in pathology even in normal mucosa. In contrast to our initial guidelines, recent studies by Lui et al. have demonstrated that the presenting symptoms of aGVHD do not significantly influence the diagnostic yield of FS and EGD [20]. Arjan et al. found a higher diagnostic yield for FS in upper abdominal symptoms and for EGD in lower ones, suggesting the necessity of performing both FS and EGD irrespective of the symptoms. Furthermore, Thompson et al.’s study (2006) supports this approach by highlighting that distal colon biopsy offers the highest diagnostic yield in patients with diarrhea at risk of aGVHD. This pivotal finding has been integrated into our formulation of the aGVHD guidelines [21].

Case 1. We present a 67-year-old male with blastic plasmacytoid dendritic cell neoplasm post 10 out of 10 matched unrelated donor Allo-SCTs. The patient was conditioned with fludarabine, melphalan, and rabbit antithymocyte globulin (ATG). Post-transplant, the patient experienced continued and worsening anorexia and odynophagia but did not complain of any diarrhea. On post-transplant day 22, the patient underwent an EGD which revealed mild inflammation of the distal esophagus, Los Angeles grade A esophagitis of the lower third of the esophagus. (Figure 1) and mild erythema of the gastric body (Figure 2). Both sites were biopsied, and the pathology report revealed a gastric fundic gland mucosa with rare epithelial apoptosis (up to one apoptotic body per tissue fragment), possible for mild acute graft-versus-host disease and a normal squamous esophageal mucosa. No viropathic changes were noted and immunostains for CMV were negative. He was started on prednisone 1 milligram per kilogram (mg/kg) daily, with a subsequent improvement in and resolution of symptoms. In this case, the EGD not only helped to confirm the diagnosis of aGVHD, but it also helped in excluding other potential viral causes of esophagitis or gastritis.

#### 2.1.2. Chronic GVHD 

Chronic GVHD (cGVHD) can occur in 20–50% of patients following Allo-SCT between 3 months and 2 years. Unlike aGVHD, cGVHD is associated with the use of peripheral blood rather than marrow as the stem cell source and in recipients of mismatched or unrelated stem cells [2]. cGVHD typically manifests as fibrosis and chronic inflammation of the skin, lungs, GIT, and soft tissues [3]. Thus, the diagnosis of cGVHD should be considered in any patient who presents with symptoms affecting these organs following Allo-SCT for more than 100 days [3]. 

Up to 60% of patients with cGVHD have some cutaneous involvement ranging from generalized erythema, plaques, and waves of desquamation, associated photoactivation, mottled pigmentation, telangiectasias, sclerotic manifestations, or joint contractures similar to scleroderma [23]. Oral mucosa dryness, mucositis, and gingivitis can also be seen [24,25]. A study by Inamoto et al. [26] found that more than two-thirds of patients with cGVHD reported eye-related symptoms, including photophobia, dryness of the eyes, and even blindness [26,27]. Genitalia may be affected in cGVHD; female patients have reported symptoms such as vaginal dryness, dyspareunia, amenorrhea, and vaginal stenosis; male patients have reported painful sexual intercourse originating from the glans penis, urethra, or meatus [28,29,30,31]. Up to half of patients with cGVHD have abnormal liver function tests [3]. 

When the GIT is affected, symptoms such as nausea/vomiting, anorexia, chronic diarrhea, malabsorption, failure to thrive, and even exocrine pancreatic insufficiency arise [32]. Although the diagnosis of cGVHD can be made based on clinical grounds, the symptoms can be nonspecific, mimicking a variety of pathologies. Endoscopy with tissue biopsy is often warranted to help assist with the diagnosis, which can reveal loss of vascular markings, focal erythema, edema, exudates, erosions, and sometimes ulceration [32]. A working group established by the haemato-oncology subgroup of the British Committee for Standards in Hematology (BCSH) and the British Society for Bone Marrow Transplantation (BSBMT) recommends that patients with diarrhea without associated jaundice or rash suggestive of cGVHD should be investigated by both upper (with duodenal aspirate and biopsies) and lower (flexible sigmoidoscopy and biopsy) GI endoscopy in preference to colonoscopy alone [33]. The use of FS is safe and appears to reduce the rate of adverse events associated with endoscopy exploring more proximal areas of the large intestine [34]. 

Similar to aGVHD, we recommend the use of EGD or FS with tissue biopsy in all patients with suspected cGVHD depending on the predominant presenting symptoms. However, for patients with isolated diarrhea without other signs of cGVHD, a combination of upper and lower endoscopy with tissue biopsies (that should be taken even in the presence of normal mucosa) may be warranted. 

Case 2. We present a 68-year-old female with a past medical history of Philadelphia-negative Chronic Lymphoblastic Leukemia (CLL) post 10/10 matched related donor Allo-SCT 8 years prior. Due to complications with cGVHD, they were presented to our institution for the evaluation of daily multiple episodes of watery diarrhea for 3 days, associated with abdominal cramping and fever. At the time of initial transplant, the patient was conditioned with rituximab, bendamustine, and fludarabine. On physical examination, she was found to have extensive skin scleroderma with mild scleral icterus. Laboratory tests revealed a cholestatic abnormality in the liver function test. cGVHD was suspected; a stool antigen panel was sent out, which was negative. FS was performed, which revealed an ulcerated mucosa in the rectum and rectosigmoid colon respectively, as seen in (Figure 3) and (Figure 4); a normal mucosal of the sigmoid colon was revealed, as seen in (Figure 5). Tissue biopsies were consistent with GVHD. She was started on a high dose of steroids and supportive care for symptomatic relief.

## 3. Hepatic Sinusoidal Obstruction Syndrome and Complications

Hepatic sinusoidal obstruction syndrome (SOS) is a systemic endothelial disease that is initiated by injury to sinusoidal endothelial cells, amplified by a local inflammatory response and the activation of coagulation and fibrinolytic pathways, causing liver necrosis in severe disease. The disease is also termed veno-occlusive disease (VOD) [35,36,37]. Conditioning regimens used pretransplant, such as alkylating agents and radiation, have been incriminated in the pathogenesis of hepatic SOS [36]. Other factors can place patients at risk of developing SOS, including liver disease prior to transplant and regimen used for GVHD prophylaxis [38,39,40]. 

Most cases of hepatic SOS occur within 4 weeks following SCT. Weight gain is usually the initial presenting symptom, followed by painful hepatomegaly and jaundice. Some patients may suffer multiorgan failure based on the severity of the disease, with renal failure being present in almost half the patients. Accompanied thrombocytopenia is often refractory to transfusions. A cholestatic pattern of liver dysfunction is usually observed, with involvement of the synthetic function in severe disease [41]. 

Hepatic SOS diagnosis can be made clinically in patients that have undergone Allo-SCT that present with signs and symptoms suggesting the pathology by using the European Society for Blood and Marrow Transplantation (EBMT) revised diagnostic criteria for adults. There are no specific biomarkers, imaging, or biopsy characteristics that can confirm the diagnosis [42]. 

Hepatic SOS, based on severity, has the potential to cause portal hypertension, resulting in collateral esophageal, gastric, and colonic varix [43]. Varices can rupture and lead to GI bleeding. Colon variceal bleeding is rare compared to esophageal and gastric bleeding; however, colonic variceal bleeding has been reported in the setting of hepatic SOS [10,44,45,46]. The only case reported in the literature of GI varices rupture in the setting of hepatic SOS was treated with esophageal variceal ligation (EVL), followed by intestinal bleeding within 1 week. The occurrence of the intestinal bleed was thought to arise from an increase in pressure in the collateral intestinal varices following the ligation of esophageal varices. The intestinal bleeding gradually resolved with somatostatin infusion and platelet transfusion. Somatostatin is effective in reducing gastric and duodenal blood flow in patients with upper GI bleeding and decreasing portal blood flow without affecting systemic blood flow [47]. 

There is a scarcity of data in the literature addressing endoscopic diagnosis and management of GI varices in the setting of hepatic SOS. As discussed above, endoscopy remains a safe technique even in the setting of thrombocytopenia that may be refractory to transfusion in hepatic SOS. Furthermore, while using EVL as endoscopic management for esophageal/gastric varices, somatostatin use should be encouraged to reduce the risk of intestinal collateral variceal rupture. Further prospective studies are needed to elucidate stronger recommendations on this subject.

## 4. CMV GI Infection and Endoscopic Diagnosis

Allo-SCT recipients are at high risk of infection that could be viral, bacterial, or even fungal that can cause serious morbidity and mortality [48]. Older age, the underlying disease for which they underwent the Allo-SCT, particularly in the setting of extensive prior therapy, prior infection of either the recipient or the donor, prior immunity to some viruses (i.e., CMV, herpes simplex virus (HSV), varicella zoster virus (VZV), and/or Epstein–Barr virus (EBV)), iron overload, myeloablative conditioning chemotherapy, and many other factors, increase the risk of post-Allo-SCT infection [49,50]. 

CMV reactivation has been reported in the early (within 100 days post-Allo-SCT) and late post-engraftment period. The CMV serostatus has been reported to significantly affect mortality; seronegative recipients transplanted with seropositive donors have a significantly higher mortality [51]. CMV infection can affect the lungs and lead to either nodular lesions or diffuse infiltrates [52,53,54,55]. CMV hepatitis should also be of consideration in the early post-engraftment period when evaluating liver function derangements [53]. CMV can also affect the bladder, and cases of post-Allo-SCT CMV hemorrhagic cystitis have been reported in the literature [56,57]. Rare cases of CMV encephalitis have also been reported in the literature [58,59,60]. 

CMV post-Allo-SCT can affect the GIT and cause symptoms with or without concomitant GVHD [7]. When the esophagus is affected, patients often complain of odynophagia, nausea, vomiting, and, in rare instances, upper GI bleeding [61,62,63]. Gastric involvement can present with epigastric pain that often improves when the patient is lying supine [64,65]. CMV colitis often presents with abdominal pain, persistent small-volume diarrhea, and rectal bleeding [66,67,68]. Rarely, toxic megacolon in the setting of CMV colitis has been reported [69]. As discussed previously, these symptoms can mimic conditioning chemotherapy mucositis and GVHD [7]. Thus, upper and/or lower GI endoscopy with tissue biopsy is warranted in patients with suspected CMV GI infection. 

On EGD, CMV esophagitis appears as a sharply demarcated serpiginous ulceration that forms giant ulcers, with occasional severe distal erosive esophagitis [62]. Although a tissue biopsy is always needed, these ulcers’ characteristics can help differentiate CMV from other forms of esophagitis, such as vesicular HSV or superficial aphthous ulcerations [70,71]. The small intestine EGD of CMV shows duodenal mucosal erythema or pseudotumors that have also been reported in the stomach [72,73,74]. Lower endoscopy for CMV colitis often shows patchy mucosal erythema with associated edema, subepithelial hemorrhage, and less commonly pseudomembranes [75,76]. 

## 5. Role of Capsule Endoscopy in GVHD

The role of capsule endoscopy (CE) in GVHD has emerged as a valuable tool in managing GVHD, offering a non-invasive technique for visualizing the small intestine. Recent studies have shown that CE detects intestinal and mucosal changes that are not detected by conventional endoscopies or radiographic imaging [77]. In a study by Smith et al., CE proved essential in monitoring the progression and response to therapy in GVHD. The study mentions that CE is useful, particularly in the long-term management of GVHD, as it is non-invasive and better tolerated by patients than conventional endoscopies. This allows for the frequent monitoring and adjustment of therapeutic agents based on mucosal healing and ongoing inflammation [78]. A study conducted by Varkey et al. showed evidence that CE had higher sensitivity in identifying or dismissing GVHD [79].

## 6. Conditioning Regimen in Hematopoietic Stem Cell Transplantation (HSCT)

It is essential to consider the impact of different conditioning regimens on patient outcomes. Conditioning regimens, categorized into myeloablative (MA), reduced-intensity conditioning (RIC), and non-MA (NMA) regimens, are crucial in preparing patients for hematopoietic stem cell transplantation (HSCT) [80,81,82]. Myeloablative regimens, typically involving high doses of chemotherapy and/or radiation, are associated with increased rates of complications, including graft-versus-host disease (GVHD), due to their aggressive nature. On the other hand, RIC regimens, which use lower doses of chemotherapy, aim to reduce toxicity while still facilitating engraftment [80]. Recent studies have highlighted that RIC regimens may result in lower rates of severe GVHD and treatment-related mortality, making them a viable option for older patients or those with comorbidities. The use of RIC for allo-HSCT has revolutionized treatment options for elderly patients and those with comorbidities. Currently, RIC is utilized in 40% of all Allo-HSCTs, and this trend continues to rise [80]. Furthermore, newer conditioning strategies, such as incorporating targeted agents and immunomodulatory drugs, are being explored to enhance the efficacy and safety of HSCT [83].

## 7. Prophylactic Measures in the Management of Post-Allogeneic Stem Cell Transplantation (Allo-SCT) Complications

Prophylactic measures against viral infections, especially cytomegalovirus (CMV), are crucial in managing post-allogeneic stem cell transplantation (Allo-SCT) complications. CMV is a common and potentially severe complication following Allo-SCT, necessitating effective prophylaxis to prevent infection and reduce the incidence of associated complications. Antiviral prophylaxis, such as ganciclovir or valganciclovir, has significantly decreased the risk of CMV reactivation and disease [83]. Recommendations for preventing opportunistic infections in hematopoietic stem cell transplant recipients include using prophylactic or preemptive ganciclovir for cytomegalovirus disease, prophylactic acyclovir for herpes simplex virus disease, fluconazole for candidiasis, and trimethoprim-sulfamethoxazole for Pneumocystis carinii pneumonia. Adhering to these guidelines aims to reduce morbidity and mortality from these infections in transplant recipients [84]. Prophylactic strategies reduce the incidence of CMV and the need for invasive diagnostic procedures, such as endoscopy, to confirm CMV-related gastrointestinal disease. Prophylactic measures against acute and chronic graft-versus-host disease (GVHD), including immunosuppressive agents and preemptive therapies, are vital in improving patient outcomes.

## 8. Post-Stem Cell Transplant Immunosuppressive Therapy

Immunosuppressive therapies are pivotal in reducing the risk of graft-versus-host disease (GVHD) following allogeneic stem cell transplantation (Allo-SCT) [85]. Agents such as cyclosporine, tacrolimus, and methotrexate are commonly used to prevent acute and chronic GVHD by suppressing the immune response that leads to tissue damage [86]. These therapies mitigate the incidence and severity of GVHD and influence the need for and outcomes of endoscopic procedures. By effectively controlling GVHD, immunosuppressive treatments reduce gastrointestinal complications, often a primary indication for endoscopy in post-transplant patients. Reducing GI complications subsequently decreases the necessity for invasive diagnostic procedures and improves overall patient outcomes.

## 9. Advancements in Regenerative Medicine

Advancements in regenerative medicine can transform patient care following allogeneic stem cell transplantation (Allo-SCT), potentially reducing the need for endoscopic procedures. Regenerative therapies, such as stem cell-based treatments and tissue engineering, aim to repair or replace damaged tissues and organs, offering novel approaches to manage complications arising from transplantation. For instance, mesenchymal stem cells (MSCs) have shown promise in reducing inflammation and promoting tissue healing, thereby mitigating the severity of gastrointestinal graft-versus-host disease (GVHD) [87]. These advancements can improve gastrointestinal health, reducing the frequency and necessity of invasive diagnostic procedures like endoscopy. Furthermore, regenerative medicine approaches could enhance the recovery and functionality of the gastrointestinal tract, contributing to better overall outcomes for transplant recipients [88].

## 10. Conclusions

Allo-SCT has shown effectiveness as a potential cure for many pathologies. With recent evidence behind the use of matched and mismatched unrelated donors in several conditions, the use of Allo-SCT is projected to increase over the years. Given the morbid nature of the procedure, several complications are associated with Allo-SCT, some of which can be diagnosed and or managed with GI endoscopy. By way of this review of the literature, we conclude that for patients with suspected aGVHD, the use of upper or lower endoscopy, to help assist the diagnosis, should be guided by the patient’s predominant GI symptom rather than systematically obtaining both for all patients. A similar strategy should be adopted for cGVHD, with the exception of patients that present with isolated GI symptoms who may require both upper and lower GI endoscopy. Tissue biopsies obtained will also differentiate GVHD with other etiologies. It is deemed to be safe to proceed with EGD and/or FS even in the setting of neutropenia and/or thrombocytopenia. Furthermore, a biopsy of the GI mucosa should always be obtained even with a normal appearing mucosa. Lastly, in the event of esophageal variceal rupture secondary to portal hypertension caused by hepatic SOS, if EVL is used for management, the patient should be concomitantly treated with somatostatin to reduce the risk of intestinal collateral varices rupture.

## 11. Selection Criteria

The databases used for this study are Cochrane, MEDLINE, Google Scholar, and PubMed.

All studies were included from PubMed, Google Scholar, and Cochrane from inception to 2024.

## 12. Limitations of the Study

In a narrative review, the inclusion of all databases is not mandatory.

As a narrative review, our paper lacked an assessment of the quality of the studies.

As a narrative review, our paper may have faced publication bias.

Standardization histopathological diagnostic criteria were not assessed.

## Figures and Tables

**Figure 1 jcm-13-04343-f001:**
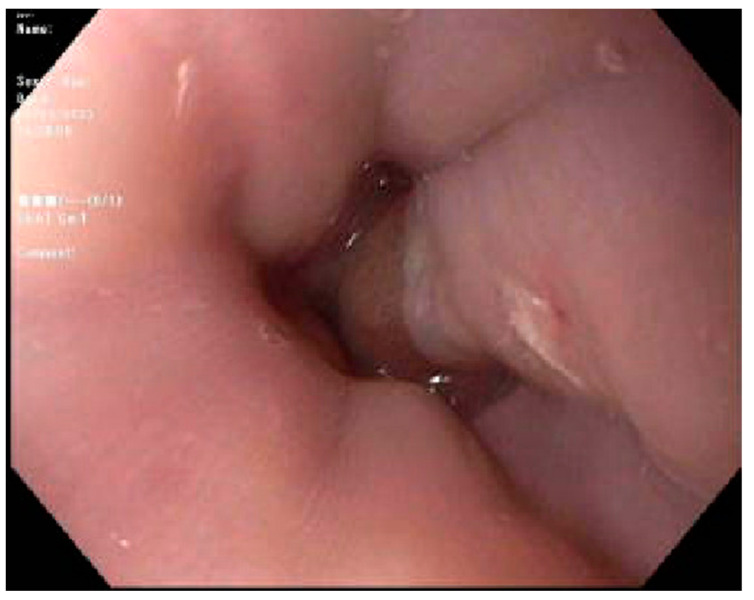
Los Angeles grade A esophagitis of the lower third of the esophagus.

**Figure 2 jcm-13-04343-f002:**
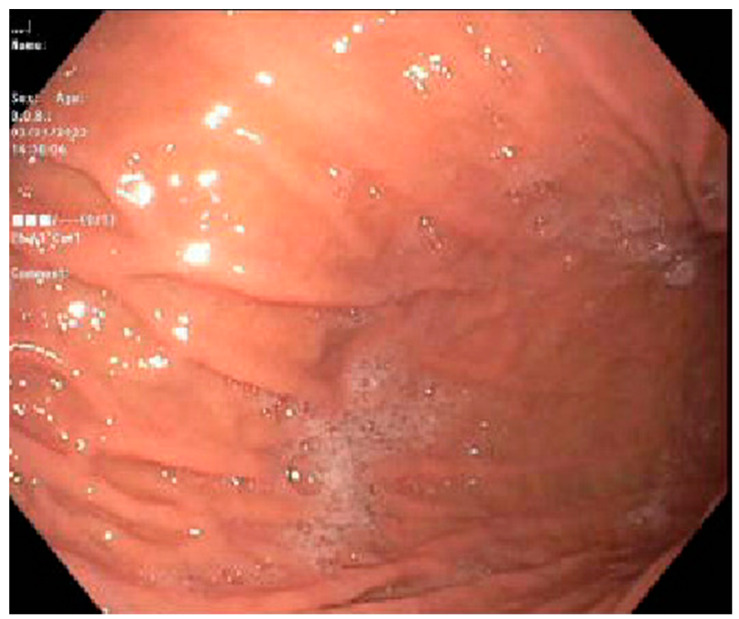
Gastric body with mild erythematous mucosa.

**Figure 3 jcm-13-04343-f003:**
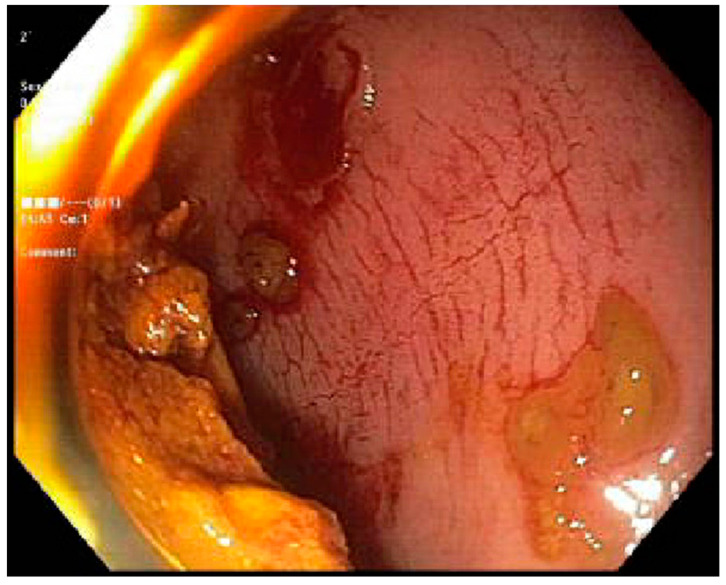
Flexible sigmoidoscopy of the rectosigmoid colon showing mucosa ulceration.

**Figure 4 jcm-13-04343-f004:**
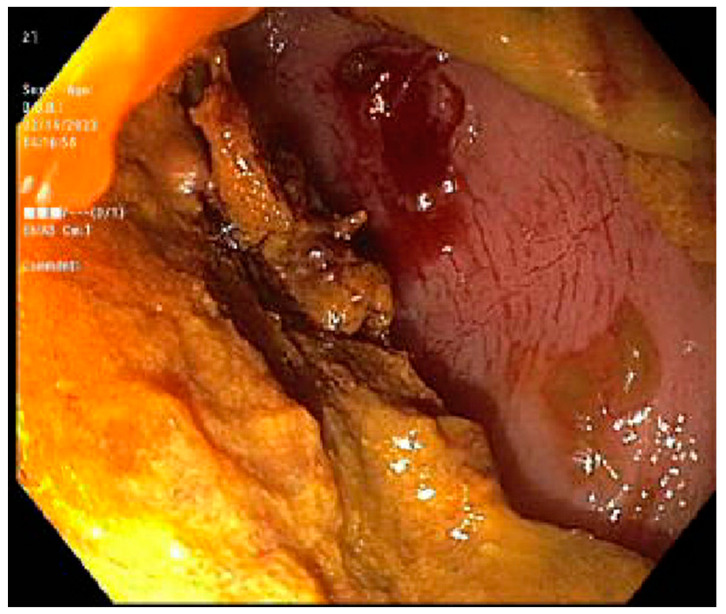
Flexible sigmoidoscopy of the rectum colon showing mucosal ulceration with stool.

**Figure 5 jcm-13-04343-f005:**
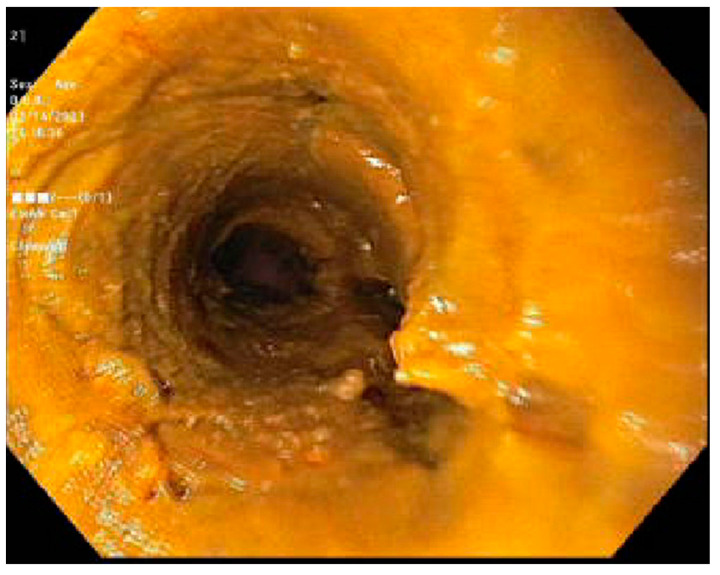
Flexible sigmoidoscopy of the sigmoid colon showing normal mucosa and poor preparation with stool throughout.

## Data Availability

All data generated or analyzed during this study are available from the corresponding author upon request.

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
