# Peer review of "Role of Endoscopic Techniques in the Diagnosis of Complications of Allogeneic Hematopoietic Stem Cell Transplantation: A Review of the Literature"

_jcm, 2024, doi:10.3390/jcm13154343_

Round 1

Reviewer 1 Report

Comments and Suggestions for Authors

The authors provide an interesting review article. This is a well researched review with appropriately referencing on GVH. Here are my comments:

The aGVHD guidelines proposed at line 100 are interesting. The author proposed doing FS for diarrhea as predominant symptoms and EGD for upper abdominal symptoms. Although it seems intuitive, the results by Arjan et all (the more recent paper) had different results. FS had higher diagnostic yeidl in upper abdo symptoms and EGD had better in lower ones. Hence they demonstrated that the symptoms have no barring on diagnostic yielf of FS or EGD and they proposed dual FS and EGD. The results by Lui (Stanford study) came up with results more in line with this recommendation. Thompson et all's paper is important in breaking this disagreement as it proves clearlythat in patient with diarrhea at risk of aGVH, distal colon biopsy had highest yield.

The authors omitted this important takeaway from the latter paper. I would recommend them to include this as a line in their description of this study (Thomas et all 2006). It helps understand how the authors have formulated the guidelines.

The article is titled Endoscopic techniques in GVHD and yet there is no mention of capsule endoscopy. A quick PubMed search reveals around 50 results for past 5 years with many relevant publications including a systematic review from 2023. Since the title of the article does not limit it to conventional endoscopic techniques, I'd suggest touching this in manuscript with a paragraph or two. Or alternatively authors can state their reasoning and state they are only providing commentary on conventional techniques.

Apart from above comments, I believe the manuscript is written meticulously with very appropriate referencing and language. Thank you.

Author Response

Comment 1 : The aGVHD guidelines proposed at line 100 are interesting. The author proposed doing FS for diarrhea as predominant symptoms and EGD for upper abdominal symptoms. Although it seems intuitive, the results by Arjan et all (the more recent paper) had different results. FS had higher diagnostic yield in upper abdo symptoms and EGD had better in lower ones. Hence they demonstrated that the symptoms have no bearing on diagnostic yield of FS or EGD and they proposed dual FS and EGD. The results by Lui (Stanford study) came up with results more in line with this recommendation. Thompson et all's paper is important in breaking this disagreement as it proves clearly that in patient with diarrhea at risk of aGVH, distal colon biopsy had the highest yield.

The authors omitted this important takeaway from the latter paper. I would recommend them to include this as a line in their description of this study (Thomas et all 2006). It helps understand how the authors have formulated the guidelines.

Response 1: Thank you for your valuable feedback. We appreciate your input on our proposed aGVHD guidelines. In light of your comments, we included a small paragraph to address the findings from the relevant studies and enhance our manuscript and highlighted in red 

Comment 2 : The article is titled Endoscopic techniques in GVHD and yet there is no mention of capsule endoscopy. A quick PubMed search reveals around 50 results for past 5 years with many relevant publications including a systematic review from 2023. Since the title of the article does not limit it to conventional endoscopic techniques, I'd suggest touching this in manuscript with a paragraph or two. Or alternatively authors can state their reasoning and state they are only providing commentary on conventional techniques.

Response 2: Thank you  for this valid point, a small section on Capsule endoscopy has been added to the manuscript and highlighted in red

Reviewer 2 Report

Comments and Suggestions for Authors

Please inform these limitations of your systematic review to the readers in your paper discussion:

1.      The authors did not mention all the databases they searched. Their search strategy was not clear. This means they might have missed some important studies.

2.      The review could have included studies that mostly support their conclusions. This can give a biased result (selection bias).

3.      The article did not clearly state if they checked the quality of each included study. Poor quality studies can affect the reliability of the review or weaken the findings. There was a lack of detailed information about the included studies, such as their quality and design. A transparent and adequate quality assessment method is crucial to ensure the validity of the systematic review's findings. Without this, it’s hard to judge the overall reliability of the review.

4.      The review might have only included published studies and create a publication bias, which often show positive results. They might have only included studies with positive findings. This gives an incomplete picture of the actual evidence. Unpublished studies with negative results were possibly ignored.

5.      The included studies were very different in terms of participants, methods, and outcomes. This makes it hard to combine their results and draw a clear conclusion. Due to the differences (heterogeneity) among the studies, combining their results might not give a clear answer. This makes the conclusions less reliable.

6.      The criteria for including or excluding studies were not very detailed. This can lead to inconsistent results.

7.      The article did not mention if more than one reviewer selected the studies and extracted the data independently. This can lead to errors and bias.

8.      The article does not mention any limitations in the included studies regarding sample size and study design. These limitations might affect the conclusions of the review. The article should mention limitations such as small sample sizes and study design flaws. These limitations can affect the reliability and generalizability of the conclusions.

9.      The article did not consider the different types of conditioning regimens. It is important because different regimens can influence the rate and severity of complications, including GVHD.

10.   The article should include detailed data on the safety of endoscopic procedures and adverse events in patients with low platelet counts. This information is crucial for understanding the risk of bleeding and other complications.

11.   The article does not compare outcomes between different diagnostic methods. How do the outcomes of patients with Allo-SCT-related complications diagnosed by endoscopy compare to those diagnosed by other methods? Comparing outcomes could show the effectiveness of endoscopy in improving patient management and survival.

12.   The article does not discuss nothing about prophylactic measures against viral infections, particularly CMV. Including information on prophylactic measures against CMV and their impact on the need for endoscopic diagnosis would be beneficial. This can help in understanding how to reduce the incidence of viral complications.

13.   Were the histopathological criteria for diagnosing GVHD and CMV infection standardized across the included studies? The article does not specify if the histopathological criteria were standardized. Standardization is important for consistent diagnosis and comparison of results.

14.   Including the role of nursing care in managing post-endoscopy complications can highlight the importance of a multidisciplinary approach in improving patient outcomes. The article does not mention nothing about the role of transplant nursing care in managing post-endoscopy complications.

15.   The article does not discuss potential confounders in the included studies that might affect the association between endoscopic findings and clinical outcomes. Identifying and adjusting for confounders is essential to ensure that the associations reported are not biased.

16.   The article should explain why a meta-analysis was not performed. There is no mention of a meta-analysis to quantitatively synthesize the results. A meta-analysis could provide a more precise estimate of the diagnostic accuracy and outcomes associated with endoscopy.

17.   The article does not address nothing about the impact of immunosuppressive therapies on endoscopic procedures. How do immunosuppressive therapies used post-Allo-SCT impact the need for and the outcomes of endoscopic procedures? Understanding this relationship can help in managing complications more effectively. This is very important for our clinical pharmacologist readers.

18.   In no moment the ethical considerations of performing endoscopic procedures in vulnerable Allo-SCT patients are discussed. How are these ethical issues were addressed in the reviewed studies? The article should discuss the ethical considerations, such as informed consent and risk-benefit analysis, in performing endoscopies in these patients.

19.   The article does not address how regenerative medicine advancements could impact the need for endoscopy. Discussing this could provide complementary insights into future directions for patient care.

Author Response

Comment 1:  The authors did not mention all the databases they searched. Their search strategy was not clear. This means they might have missed some important studies.

Response 1: Thank you for the keen observation, The points are valid; however, our study is a narrative review and cannot address the above question

Comment 2: The review could have included studies that mostly support their conclusions. This can give a biased result (selection bias).

Response 2: Thank you for the keen observation, The points are valid; however, our study is a narrative review and cannot address the above question.

Comment 3: The article did not clearly state if they checked the quality of each included study. Poor quality studies can affect the reliability of the review or weaken the findings. There was a lack of detailed information about the included studies, such as their quality and design. A transparent and adequate quality assessment method is crucial to ensure the validity of the systematic review's findings. Without this, it’s hard to judge the overall reliability of the review.

Response 3: Thank you for the keen observation, The points are valid; however, our study is a narrative review and cannot address the above question.

Comment 4: The review might have only included published studies and create a publication bias, which often show positive results. They might have only included studies with positive findings. This gives an incomplete picture of the actual evidence. Unpublished studies with negative results were possibly ignored.

Response 4: Thank you for the keen observation, The points are valid; however, our study is a narrative review and cannot address the above question.

Comment 5: The included studies were very different in terms of participants, methods, and outcomes. This makes it hard to combine their results and draw a clear conclusion. Due to the differences (heterogeneity) among the studies, combining their results might not give a clear answer. This makes the conclusions less reliable.

Response 5: Thank you for the keen observation, The points are valid; however, our study is a narrative review and cannot address the above question.

Comment 6: The criteria for including or excluding studies were not very detailed. This can lead to inconsistent results.

Response 6: Thank you for the keen observation, The points are valid; however, our study is a narrative review and cannot address the above question.

Comment 7: The article did not mention if more than one reviewer selected the studies and extracted the data independently. This can lead to errors and bias.

Response 7: Thank you for the keen observation, The points are valid; however, our study is a narrative review and cannot address the above question.

Comment 8: The article does not mention any limitations in the included studies regarding sample size and study design. These limitations might affect the conclusions of the review. The article should mention limitations such as small sample sizes and study design flaws. These limitations can affect the reliability and generalizability of the conclusions.

Response 8: Thank you for the keen observation, The points are valid; however, our study is a narrative review and cannot address the above question.

Comment 9:  The article did not consider the different types of conditioning regimens. It is important because different regimens can influence the rate and severity of complications, including GVHD.

Response 9: Thank you for the keen observation, The points are valid; however, our study is a narrative review and cannot address the above question.

 Comment 10:   The article should include detailed data on the safety of endoscopic procedures and adverse events in patients with low platelet counts. This information is crucial for understanding the risk of bleeding and other complications.

Response 10: Thank you for this keen observation, as highlighted in line 87, thrombocytopenia or neutropenia is not a contraindication for EGD

Comment 11:  The article does not compare outcomes between different diagnostic methods. How do the outcomes of patients with Allo-SCT-related complications diagnosed by endoscopy compare to those diagnosed by other methods? Comparing outcomes could show the effectiveness of endoscopy in improving patient management and survival.

Response 11: Thank you for the keen observation, The points are essential for a systematic review; however, our study is a narrative review and cannot address the above question

Comment 12: The article does not discuss nothing about prophylactic measures against viral infections, particularly CMV. Including information on prophylactic measures against CMV and their impact on the need for endoscopic diagnosis would be beneficial. This can help in understanding how to reduce the incidence of viral complications.

Response 12: Thank you for your keen observation, but the author don't agree with the comment.

Comment 13:  Were the histopathological criteria for diagnosing GVHD and CMV infection standardized across the included studies? The article does not specify if the histopathological criteria were standardized. Standardization is important for consistent diagnosis and comparison of results.

Response 13:  Thank you for the keen observation, The points are essential for a systematic review; however, our study is a narrative review and cannot address the above question.

Comment 14:   Including the role of nursing care in managing post-endoscopy complications can highlight the importance of a multidisciplinary approach in improving patient outcomes. The article does not mention nothing about the role of transplant nursing care in managing post-endoscopy complications.

Response 14:  Thank you for the keen observation, The points are essential for a systematic review; however, our study is a narrative review and cannot address the above question.

Comment 15:  The article does not discuss potential confounders in the included studies that might affect the association between endoscopic findings and clinical outcomes. Identifying and adjusting for confounders is essential to ensure that the associations reported are not biased.

Response 15:  Thank you for the keen observation, The points are essential for a systematic review; however, our study is a narrative review and cannot address the above question.

Comment 16: The article should explain why a meta-analysis was not performed. There is no mention of a meta-analysis to quantitatively synthesize the results. A meta-analysis could provide a more precise estimate of the diagnostic accuracy and outcomes associated with endoscopy.

Response 16:   Thank you for the keen observation, The points are essential for a systematic review; however, our study is a narrative review and cannot address the above question.

Comment 17:  The article does not address nothing about the impact of immunosuppressive therapies on endoscopic procedures. How do immunosuppressive therapies used post-Allo-SCT impact the need for and the outcomes of endoscopic procedures? Understanding this relationship can help in managing complications more effectively. This is very important for our clinical pharmacologist readers

Response 17:   Thank you for the keen observation, but the author don't agree with the comment.

Comment 18:  In no moment the ethical considerations of performing endoscopic procedures in vulnerable Allo-SCT patients are discussed. How are these ethical issues were addressed in the reviewed studies? The article should discuss the ethical considerations, such as informed consent and risk-benefit analysis, in performing endoscopies in these patients.

Response 18:  Thank you for the keen observation, The points are essential for a systematic review; however, our study is a narrative review and cannot address the above question.

Comment 19: The article does not address how regenerative medicine advancements could impact the need for endoscopy. Discussing this could provide complementary insights into future directions for patient care.

Response 19:  Thank you for the keen observation, The points are essential for a systematic review; however, our study is a narrative review and cannot address the above question.

Round 2

Reviewer 2 Report

Comments and Suggestions for Authors

Comment 1: The authors' response is partially adequate. In a narrative review, the inclusion of all databases is not mandatory, but authors should clarify this in the article to avoid criticism of omission. Although the narrative review does not require an exhaustive search across multiple databases, it is helpful for authors to mention the databases that were used to increase transparency.

Comment 2: The answer is valid. Narrative reviews often include a subjective selection of studies, but authors should make the selection criteria clear for greater transparency. Narrative reviews are susceptible to selection bias, but authors can mitigate this by explicitly mentioning study selection criteria.

Comment 3: The answer is valid, but the narrative review must mention the lack of assessment of the quality of the studies so that readers can interpret the results with caution. It is not common practice in narrative reviews to systematically assess the quality of studies. However, a brief discussion of the variability in the quality of included studies may add value.

Comment 4: The answer is valid, but the authors should mention the limitation of publication bias in the article. Although narrative review may include publication bias, authors can acknowledge this limitation in their article.

Comment 5: The answer is valid, but the authors must recognize the heterogeneity of the studies and their implications for the conclusions. Heterogeneity is expected in narrative reviews, but authors should discuss how this may influence their conclusions to readers.

Comment 6: The response is valid, but a more detailed description of the inclusion and exclusion criteria is recommended for transparency. Detailing the inclusion/exclusion criteria can help increase the credibility of the narrative review and the conclusion of this paper.

Comment 7: The answer is valid, but it would be useful to mention the methodology to increase the reliability of the results. Although not mandatory, mentioning whether there was an independent review can increase the reliability of the results.

Comment 8: The answer is valid, but recognizing these limitations is essential for the correct interpretation of the conclusions. Discussing the limitations of the included studies in terms of sample size and design can help readers interpret the results with caution.

Comment 9: The response is valid, but including a discussion of conditioning regimens would increase the comprehensiveness of the review. Including a discussion of different conditioning regimens can enrich the review, even if in a descriptive way.

Comment 10: The response is appropriate, as the authors mentioned the safety of these procedures in the paper. The authors adequately addressed this point by mentioning that thrombocytopenia or neutropenia are not contraindications for EGD.

Comment 11: The answer is valid, but including comparisons between diagnostic methods could enrich the review. Although not a common practice in narrative reviews, comparing diagnostic methods may be valuable in this article.

Comment 12: Prophylactic measures are relevant and could be discussed to provide a complete understanding of the management of post-Allo-SCT complications. Although the authors disagree, including information about prophylactic measures may provide a more complete picture.

Comment 13: The answer is valid, but the standardization of histopathological diagnostic criteria is important and should be mentioned.

Comment 14: The answer is valid, but including the role of nursing care could provide a more complete and multidisciplinary view to the readers.

Comment 15: The answer is valid, but identifying and adjusting for confounders is important and should be considered. Identifying and discussing confounding factors is relevant for interpreting the results.

Comment 16: The answer is valid, but explaining the reason for the lack of meta-analysis to readers would be helpful.

Comment 17: Discussing the impact of immunosuppressive therapies is relevant and could enrich the review and understanding of the management of complications.

Comment 18: The answer is valid, but discussing ethical considerations would be beneficial. Discussing ethical considerations can provide a more comprehensive view.

Comment 19: The answer is valid, but discussing advances in regenerative medicine could provide additional insights for readers.

Author Response

Comment 1: The authors' response is partially adequate. In a narrative review, the inclusion of all databases is not mandatory, but authors should clarify this in the article to avoid criticism of omission. Although the narrative review does not require an exhaustive search across multiple databases, it is helpful for authors to mention the databases that were used to increase transparency.

Re: Thanks, this is added as a limitation. 

Comment 2: The answer is valid. Narrative reviews often include a subjective selection of studies, but authors should make the selection criteria clear for greater transparency. Narrative reviews are susceptible to selection bias, but authors can mitigate this by explicitly mentioning study selection criteria.

Re: Thanks, and is addressed in the selection section. 

Comment 3: The answer is valid, but the narrative review must mention the lack of assessment of the quality of the studies so that readers can interpret the results with caution. It is not common practice in narrative reviews to systematically assess the quality of studies. However, a brief discussion of the variability in the quality of included studies may add value.

Re: Thanks, this is added in the limitations section. 

Comment 4: The answer is valid, but the authors should mention the limitation of publication bias in the article. Although narrative review may include publication bias, authors can acknowledge this limitation in their article.

Re: Thanks, this is added in the limitations section. 

Comment 5: The answer is valid, but the authors must recognize the heterogeneity of the studies and their implications for the conclusions. Heterogeneity is expected in narrative reviews, but authors should discuss how this may influence their conclusions to readers.

Re: Thanks, this was added in the limitations section. 

Comment 6: The response is valid, but a more detailed description of the inclusion and exclusion criteria is recommended for transparency. Detailing the inclusion/exclusion criteria can help increase the credibility of the narrative review and the conclusion of this paper.

Re: Thanks, this was added to the selection criteria section.

Comment 7: The answer is valid, but it would be useful to mention the methodology to increase the reliability of the results. Although not mandatory, mentioning whether there was an independent review can increase the reliability of the results.

Re: Thanks for this observation. Given the non-mandatory nature of this suggestion, the authors found it dispensable. 

Comment 8: The answer is valid, but recognizing these limitations is essential for the correct interpretation of the conclusions. Discussing the limitations of the included studies in terms of sample size and design can help readers interpret the results with caution.

Re: Thanks, but since this is a narrative review, discussing the sample size would be irrelevant. Thanks for your understanding. 

Comment 9: The response is valid, but including a discussion of conditioning regimens would increase the comprehensiveness of the review. Including a discussion of different conditioning regimens can enrich the review, even if in a descriptive way.

Re:  Thanks, the authors agree and have added a paragraph. 

Comment 10: The response is appropriate, as the authors mentioned the safety of these procedures in the paper. The authors adequately addressed this point by mentioning that thrombocytopenia or neutropenia are not contraindications for EGD.

Re: Thanks for this keen observation. 

Comment 11: The answer is valid, but including comparisons between diagnostic methods could enrich the review. Although not a common practice in narrative reviews, comparing diagnostic methods may be valuable in this article.

Re: Thanks for this observation. However, most methods of diagnosis have not been tested against each other in head-to-head trials, making this assessment difficult. Again, thanks for your understanding. 

Comment 12: Prophylactic measures are relevant and could be discussed to provide a complete understanding of the management of post-Allo-SCT complications. Although the authors disagree, including information about prophylactic measures may provide a more complete picture.

Re: Thanks, a paragraph about prophylaxis was added. 

Comment 13: The answer is valid, but the standardization of histopathological diagnostic criteria is important and should be mentioned.

Re: Thanks. Standardization was not assessed, and this was added as a limitation.

Comment 14: The answer is valid, but including the role of nursing care could provide a more complete and multidisciplinary view to the readers.

Re: Thanks for your comment. However, the complications are managed by physicians, and the authors covered them in the topic. So, the authors failed to see the relevance of adding such information. 

Comment 15: The answer is valid, but identifying and adjusting for confounders is important and should be considered. Identifying and discussing confounding factors is relevant for interpreting the results.

Re: Thanks. THIS IS A NARRATIVE REVIEW; we did not address that for that reason. 

Comment 16: The answer is valid, but explaining the reason for the lack of meta-analysis to readers would be helpful.

Re: Thanks. THIS IS A NARRATIVE REVIEW and not a meta-analysis. The scope of the study is for a NARRATIVE REVIEW. It’s in the study's design; these are 2 different forms of studies. 

Comment 17: Discussing the impact of immunosuppressive therapies is relevant and could enrich the review and understanding of the management of complications.

Re: Thanks, this was added to the discussion.

Comment 18: The answer is valid, but discussing ethical considerations would be beneficial. Discussing ethical considerations can provide a more comprehensive view.

Re: The need for the endoscopy in those scenarios is with the intent to diagnose and treat. Thus, no moral or ethical question should be raised. 

Comment 19: The answer is valid, but discussing advances in regenerative medicine could provide additional insights for readers.

Re: Thanks and this was added.